# Does Early Childhood Caries Increase Caries Development among School Children and Adolescents? A Systematic Review and Meta-Analysis

**DOI:** 10.3390/ijerph192013459

**Published:** 2022-10-18

**Authors:** Phoebe Pui Ying Lam, Helene Chua, Manikandan Ekambaram, Edward Chin Man Lo, Cynthia Kar Yung Yiu

**Affiliations:** 1Paediatric Dentistry, Faculty of Dentistry, The University of Hong Kong, Hong Kong 999077, China; 2Auckland District Health Board, Auckland 1051, New Zealand; 3Paediatric Dentistry, Faculty of Dentistry, University of Otago, Dunedin 9016, New Zealand; 4Applied Oral Sciences & Community Dental Care, Faculty of Dentistry, The University of Hong Kong, Hong Kong 999077, China

**Keywords:** Early Childhood Caries, dental caries, adolescents, social inequality, systematic review

## Abstract

The aim of this paper is to systematically review the literature to determine whether early childhood caries (ECC) is significantly associated with caries development in permanent teeth among school children and adolescents, and to identify the association of other risk factors over 24 months. A systematic literature search was performed in four electronic databases and via a manual search from inception to 28 July 2022. Independent study selection and screening, data extraction, evaluation of risk of bias using ROBINS-I tool and certainty of evidence with GRADE were performed. Ten cohort studies were included, all of which identified that ECC significantly increased the risk of caries in permanent teeth. Meta-analysis suggested children with ECC were three times more likely to develop caries in their permanent teeth (OR, 3.22; 95% CI 2.80, 3.71; *p* < 0.001), especially when the lesions were in primary molars and progressed to dentine. However, the certainty of evidence was substantially compromised by serious risk of bias and inconsistency between studies. There were inconsistent findings between socioeconomic or behavioural factors on caries development, which could not be pooled for meta-analyses. ECC significantly increases the likelihood of caries development in permanent teeth. Evidence on the association of socioeconomic and oral health behavioural factors is weak.

## 1. Introduction

Early childhood caries (ECC) refers to the presence of decay in any primary teeth in a child aged below 72 months [1]. ECC was prevalent in around half of children globally [2], with the prevalence ranging from 2.1% to 85.5% in different developed and developing countries [3]. Unfortunately, ECC usually remained untreated [3,4]. In 2015, over 620 million children worldwide had untreated ECC [4], and among them one- to four-year-old children were the most affected [4].

Untreated ECC can lead to profound and long-lasting impacts on children [5,6]. The pain and infection resulting from untreated ECC can significantly hinder children’s schooling, sleep habits and other daily activities [5]. Long-term repercussions include diminished growth and body weight, compromised general health and poorer quality of life [5,6].

Although primary teeth would be replaced by permanent successors, ECC was suggested to be a potential risk factor for caries in permanent dentition [7]. Many epidemiological studies have suggested increased incidence and severity of caries in the permanent teeth among schoolchildren, adolescents and adults with a known history of ECC [8,9,10]. The potential explanations include the high prevalence of cariogenic bacteria in the primary dentition may have a spillover effect on the newly erupted teeth [11,12]. Another possible cause would be common risk factors that affect both dentitions, such as sociodemographic factors, oral hygiene maintenance and dietary habits that have not been modified from childhood to adolescence [13,14].

A number of systematic reviews have investigated different caries risk factors for various age groups [15,16,17,18]. Identified factors could be mostly categorized into sociodemographic, medical conditions, dietary, oral hygiene habits and oral microflora, etc. [15,16,17,18]. While some have stricter selection criteria to include only cohort and case–control studies [16,17], many reviews have included cross-sectional studies [15,18]. As dental caries would normally take more than two years to manifest clinically [19], more concrete evidence should be drawn from studies with longer review time.

As different etiological factors would require different target-specific approaches to eradicate the root causes, it is important to identify and understand the potential risk factors so that more cost-effective interventions can be implemented to prevent caries in the permanent dentition. The aim of this systematic review was to evaluate whether ECC is associated with caries development in permanent teeth among children and adolescents, and to identify the association of other risk factors with caries increment in the permanent teeth over a follow-up period of at least 24 months. 

## 2. Materials and Methods

The systematic review and meta-analyses were conducted and reported in accordance to the guidelines stated in the Preferred Reporting Items for Systematic Reviews and Meta-Analyses (PRISMA) statement [20], and Meta-Analyses of Observational Studies in Epidemiology (MOOSE) guideline [21]. The protocol of the review was registered in PROSPERO prior to commencement (registration number: CRD42021265270).

### 2.1. Information Sources and Literature Search

Four electronic databases were scrutinized with broad keywords and MeSH terms (PubMed; Ovid Medline; Ovid Embase; Web of Science) (Appendix A). Grey literature (www.opengrey.eu, accessed on 14 September 2022) and Google Scholar were searched to identify any unpublished relevant material. The reference lists of relevant clinical trials, previous reviews and included studies were also manually searched.

### 2.2. Study Selection

Two authors (first and second authors) independently evaluated reports’ titles, keywords and abstracts before deciding on their potential eligibility. Agreements between reviewers were determined with Cohen’s kappa coefficient (κ). Disagreement was resolved by consensus or by consulting the third reviewer (third author).

### 2.3. Types of Studies

Prospective and retrospective cohort or case–control studies with at least 24 months of follow-up were included. The included studies evaluated how dental caries increment in permanent teeth was associated with different potential risk factors in terms of caries prevalence, experience and incidence. Studies that examined participants clinically only once, interventional trials and reviews were excluded, but their reference lists were screened for potentially eligible studies.

### 2.4. Types of Participants

School children and adolescents (between 6 to 18 years old) with mixed or permanent dentitions, who had been examined at least once at a baseline below 72 months old, were included. Studies including children with physical or psychological disabilities were excluded. Studies including participants beyond the age range were also excluded, as caries development in adults might be subjected to other modulating factors—for instance, smoking and root caries as a result of periodontal diseases [22].

### 2.5. Types of Exposure and Control/Comparison

This systematic review evaluated whether caries in the primary dentition would increase the risk of caries in the permanent teeth. Additionally, the review also evaluated whether other risk factors, including parental socioeconomic position, dietary and oral health behavioural habits and other oral conditions might be associated with caries in the permanent teeth.

### 2.6. Measures of Effect

Dichotomous outcomes, such as the incidence of caries, were evaluated using odds ratios (OR) and 95% confidence intervals (95% CI). Continuous outcomes, such as mean amount of decay, missing teeth due to decay and filled permanent teeth (DMFT) were evaluated using standardized mean difference (SMD) and standard deviation.

### 2.7. Types of Outcome Measures

The primary outcome was the increment of dental caries in permanent teeth in terms of the decay, missing and filled teeth or surfaces (DMFT or DMFS) index (World Health Organization), or the International Caries Detection and Assessment System (ICDAS) scores. The secondary outcome was the proportion of children developing new caries in their permanent dentition.

### 2.8. Data Collection and Measurement of Treatment Effect

Two authors (first and second authors) used a standardized data extraction spreadsheet to extract relevant data independently, including study characteristics (design, year of commencement and duration), participants (location, inclusion and exclusion criteria, gender, age, baseline caries), exposure and control (parental and maternal education background, household income).

### 2.9. Risk of Bias in Individual Studies

The validity of each study was evaluated by the risk of bias in non-randomized studies of interventions (ROBINS-I) tool [23,24]. The tool consists of 7 domains, including (I) bias due to confounding, (II) bias in selection of participants into the study, (III) bias in classification of exposures, (IV) bias due to departures from intended exposures, (V) bias due to missing data, (VI) bias in measurement of outcomes and (VII) bias in selection of the reported result. Each study was determined as being of low, moderate, serious or critical risk of bias based on the result in each domain [23,24].

### 2.10. Data Synthesis & Analyses

The meta-analyses were conducted with Stata version 13.1 (StataCorp, College Station, TX, USA, 2013), with a fixed-effects model when there were fewer than 5 studies, and a random-effects model when there were 6 or more studies [25,26]. The results were reported narratively if results between studies were significantly divergent. Subgroup analyses based on confounders, and sensitivity analyses was performed if there were sufficient studies available [25]. Potential sources of heterogeneity, including risk of bias based on ROBINS-I, sample size and income distribution (Gini coefficient) of the study country, were controlled by subgroup analyses.

### 2.11. Assessment of Heterogeneity and Publication Bias

Heterogeneity between studies was evaluated using I^2^ statistics and a Chi-square test [25]. Heterogeneity was considered substantial if I^2^ was greater than 50% and the *p*-value of the Chi-square test was less than 0.05 [25]. Funnel plots were used to assess publication bias if there were more than 10 studies in each meta-analysis [26,27].

### 2.12. Assessment of Certainty of Evidence

The certainty of evidence was rated based on the guidelines suggested by the Grading of Recommendations Assessment Development and Evaluation (GRADE) approach [24]. The certainty of evidence was downgraded if there were serious concerns with respect to risk of bias, imprecision, inconsistency, indirectness and publication bias; or upgraded if there was large magnitude of effect, dose response or no plausible confounding.

## 3. Results

### 3.1. Selection Process

Figure 1 details the screening process in accordance with the PRISMA guidelines. A systematic literature search of records identified 1198 records after removing duplicates. A total of 32 articles were retrieved for full text reading and evaluation. Ten articles were found to be eligible and included for qualitative synthesis [8,9,10,28,29,30,31,32,33,34] (Kappa κ: 0.899). The list of excluded studies and the reasons for exclusion are shown in Appendix A.

### 3.2. Data Extraction

Nine prospective and one retrospective studies were included, with 7580 children ranging from 1.5 years old to 5 years old at baseline with primary dentition, and followed up until 6 to 14 years old in mixed dentition [8,9,10,28,29,30,31,32,33,34]. The duration of follow-up ranged from 3 to 12 years with 6786 subjects. These included studies that were conducted in the USA (*n* = 2) [28,32], Europe (*n* = 4) [9,30,31,33], Asia (*n* = 3) [10,29,34] and Brazil (*n* = 1) [8] (Table 1).

### 3.3. Risk of Bias of Included Studies

Figure 2 and Appendix A detail the risk of bias assessment of each study using the ROBINS-I tool. Only one study was graded as of low risk of overall bias, as it was graded as low risk of bias in all seven domains [33]. Seven studies were graded as of serious risk of overall bias, as they were graded as serious risk of bias in one or more domains [28,29,30,31,32,34,35]. The remaining studies were graded as of moderate risk of overall bias, with more than one domain being graded as of moderate risk of bias [8,9]. Hence, the findings reported in this systematic review have to be interpreted with much caution.

### 3.4. Potential Factors Associated with Caries Increment in Permanent Teeth

#### 3.4.1. Caries in Primary Dentition

All included studies consistently identified ECC as a significant risk predictor for caries in their permanent teeth [8,9,10,28,29,30,31,32,33,34]. However, only three studies reported the mean DMFT/DMFS scores in the exposure and control groups that allowed data pooling and quantitative analyses [9,28,29]. Meta-analyses from these three studies suggested children with ECC are 3.22 times (OR, 3.22; 95% CI 2.80, 3.71; *p* = 0.001) as likely to have caries in their permanent teeth [9,28,29] (Figure 3) than those without caries in their primary teeth.

#### 3.4.2. Depth of Caries and Severity of ECC

Only one study investigated whether the depth of caries in the primary dentition would post a significant effect on the caries in the permanent teeth [9]. Saethre-Sundli et al. (2015) reported that preschool children with enamel caries and dentinal caries in their primary dentitions were 1.6 (OR, 1.6, 95% CI, 1.2, 2.0, *p* < 0.05) and 3.2 (OR, 3.2, 95% CI 2.6, 3.9, *p* < 0.05) times as likely to develop dentinal caries in their permanent teeth (Figure 3) [9]. 

Du et al. (2017), on the other hand, investigated whether DMFT score was positively correlated with the likelihood of developing caries in the permanent dentition [29]. The risk ratio of caries prevalence in permanent dentition increased with increased DMFT score in primary dentition. The risk ratio increased 1.5 times (95% CI, 1.3, 1.8, *p* < 0.001) when the DMFT score was 1–3; and 15.2 times (95% CI, 7.5, 30.8, *p* < 0.001) when the DMFT score was over 13, compared to children with no caries in their primary teeth.

The severity of ECC is also positively associated with caries in permanent dentition. Children with a higher DMFS score were more likely to have caries in their permanent dentition [10]. Likewise, children with caries that progressed to dentine in their primary teeth were more likely to have caries in their permanent dentition [9] compared with children with no caries or enamel caries in the primary dentition [9].

#### 3.4.3. Location of Caries

Li & Wang (2002) investigated different primary teeth and determined each of their predictive value for caries development in permanent teeth [10]. Caries in first and second primary molars were the most accurate predictors of caries development in permanent teeth (RR, 3.4; 95% CI, 1.8, 6.1, *p* < 0.001). On the other hand, caries in upper incisors had the lowest predictive value (RR, 1.6, 95% CI, 1.07, 2.45, *p* = 0.07). However, Al-Shahan et al. (1997) showed that carious lesion in primary incisors was sufficient as a marker for predicting caries in the permanent dentition [28].

#### 3.4.4. Gender

No studies suggested gender as a significant factor associated with increased caries risk in permanent teeth [9,10,29]. As shown in the meta-analyses, boys and girls carried similar risk of developing caries in their permanent teeth (OR, 0.86; 95% CI 0.71, 1.02; *p* = 0.670) [10,29] (Figure 4).

#### 3.4.5. Sociodemographic Factors

Meta-analysis could not be performed as data could not be extracted from reported studies. Three of the included studies investigated whether sociodemographic factors were associated with increased caries increment in permanent teeth [8,9,10].

Saethre-Sundli et al. (2020) recognized single-parent family and low parental education level as significant risk factors for developing caries in permanent teeth [9]. Surprisingly, Li & Wang (2002) identified children with higher socio-economic status (SES) had significantly higher DMFT scores rather than vice versa [10].

However, in contrary to Li & Wang (2002) [10]. and Saethre-Sundli et al. (2020) [9], Cortellazzi et al. (2013) did not identify monthly family income, paternal and maternal education as being significant factors after adjusted for past caries experience [8].

#### 3.4.6. Oral Hygiene and Dietary Habits

Only two studies recorded and compared whether different oral hygiene and dietary habits were associated with future caries development [33,34]. Alm et al. (2012) found that frequent consumption of sweets (more than once per week) at one and three years old significantly increased the prevalence and severity of caries at 15 years old [33]. Chankanka et al. (2011) also identified processed sugary snacks as a significant risk factor, but not in the final logistic regression model after accounting for past caries experience [34].

For toothbrushing frequency, Chankanka et al. (2011) found lower daily toothbrushing frequency at 5–8 years old significantly increased the risk of new cavitated caries at 9 years old [34]. Alm et al. (2011) categorized toothbrushing once per day or less and consuming snacks over 14 times per week as unfavourable behaviours [33]. They found that children who demonstrated such unfavourable behaviour at 3 years old were significantly associated with caries at 15 years old [33]. However, these unfavourable behaviours were not analysed in the final regression analyses with other potential confounding factors, such as past caries experience and socioeconomic factors [33].

#### 3.4.7. Bacterial Load

Two studies investigated whether colonization of Streptococci mutans in primary dentition would affect the caries risk in the mixed and permanent dentition [31,32]. A significantly higher DMFT/DMFS score was found among school children 6 to 12 years old who had a higher Streptococci mutans load at 2–3 years old [31,32].

### 3.5. GRADE Assessment

The certainty of evidence was rated very low for ECC as an increased risk factor for caries in the permanent teeth among children. The certainty of evidence was downgraded due to observational studies, serious risk of bias of included studies and substantial heterogeneity (Table 2).

GRADE assessment was also carried out for the outcome comparison between boys and girls, as only data for gender were available for meta-analysis. The certainty of evidence for no association between gender and caries increment in permanent teeth was rated as low, as data were generated from observational studies despite no serious concerns in all five domains.

## 4. Discussion

Our findings suggest that individuals with ECC are still three times more likely to develop caries in their permanent teeth. Although primary teeth will exfoliate and be succeeded by a new set of teeth, past caries experience has been a reliable indicator for a person’s future caries risk [1]. However, the effect estimate generated should be interpreted cautiously, as all included studies in the meta-analysis had moderate or severe risk of bias due to confounding and severe overall bias. Despite the fact that most of the included studies were assigned moderate to serious risk of bias, indicating that the true effect may be affected by confounding factors or other means of bias, all included studies found that caries in primary dentition were sufficient as a standalone marker to identify the group of children most at risk of subsequent caries. Therefore, this is a serious-risk group that requires prudent attention. 

Our findings also affirm that there is a negative long-term impact of cariogenic bacteria acquired early in infancy or preschool age on the permanent teeth. Hence, preventing ECC and reducing transmission of cariogenic bacteria to young children are of paramount importance to minimize future caries development in the permanent teeth. An infant’s mouth is theoretically sterile at birth, but rapidly becomes colonized by the microbiota available. Vertical transmission from the saliva of the parents is the chief vehicle [36]. Therefore, children are more susceptible to ECC if their caregivers, especially the mother, harbour more cariogenic bacteria in saliva due to untreated dental decay [36]. Children can also acquire oral microbiota via horizontal transmission from their siblings and peers, such as when sharing foods [37].

Caries lesion depth and location of ECC can also affect caries development in the permanent teeth. This is not surprising, as more advanced caries lesions retain plaque and harbour more aciduric cariogenic bacteria, such as *Lactobacilli* and *Propionibacterium* [38], promoting caries progression at that site and in the whole oral cavity [38]. It has been reported that cavities in primary molars result in accumulation of more aciduric and cariogenic plaque next to the neighbouring newly erupted permanent molars [10].

It might be difficult to perform operative treatments among young preschool children with established lesions; hence, minimal intervention dentistry which aims to manage the biological process of dental decay should be considered [39]. For fissure and interproximal caries in primary molars, retentive fissure sealant and stainless-steel crowns placed by Hall technique can be used, respectively [40,41]. Both treatments aim to seal the cariogenic bacteria from obtaining nutrition supply and were found to be effective in arresting the progression of carious lesions [39,40,41].

Another factor that affects caries development in the permanent teeth is the oral health habits of the child. Previous studies found that the cariogenic bacterial count dropped significantly immediately after restorative treatment [42,43]. However, the cariogenic bacterial loads gradually relapsed to pretreatment level if the detrimental oral health habits were not modified [42,43]. Even if all caries lesions in primary teeth were treated or restored, the likelihood of these children having caries in their newly erupted permanent teeth was also higher than it was for those who were caries-free in their primary dentition [44,45]. Therefore, restoring decayed primary teeth alone is not a panacea to leveling prevalent dental inequalities and preventing caries. 

As most of the socioeconomic factors are not easily modifiable, more targeted oral health promotional campaign and implementation of preventive measures are required to reach the less privileged individuals, who are most at risk of oral diseases [14]. Fissure sealants and professionally applied topical fluoride agents have proven to be effective in preventing caries [46,47], with school-based and community-based preventive programs showing promising results in reducing the oral health disparity [40,48]. Oral health education and promotional programs should be implemented early, as soon as immediately after birth [49], so that healthy oral health habits can be established and continued throughout one’s lifetime.

Based on the risk of bias assessment evaluated with ROBINS-I, most of the included studies were rated as of moderate to severe risk of bias. ROBINS-I is a risk-of-bias assessment tool which allows reviewers to comprehensively and systematically analyse the included studies based on the specific guided questions in each domain [50]. However, compared with other assessment tools that evaluate the risk of bias of non-randomised studies [51], ROBINS-I has stricter requirements and often underscores the quality of the included studies with inadequate reporting [50]. Nevertheless, the included studies were rated as of moderate to severe risk of bias, as many had not taken potential confounders of caries development (including socioeconomic factors and parental education) into account. Hence, the current findings should be interpreted with caution.

Strengths of this review include dual independent selection and evaluation of the literature, risk of bias assessment and evidence assessment with GRADE approach, as the review process followed the PRISMA guideline tightly [20]. Limitations of this review include the inevitable exclusion of non-English reports, inability to carry out meta-analyses for other potential confounding factors and funnel plots due to limited number of longitudinal studies found.

## 5. Conclusions

Children with early childhood caries are more likely to develop caries in their permanent teeth, with caries specifically in the primary molars and dentinal caries further elevating the risk. Likewise, there is a positive association between the number of carious lesions in the primary dentition and the risk of caries in their permanent teeth. However, the certainty of evidence was very low due to the high risk of bias of most of the included studies. The evidence of the association between socioeconomic, oral health behavioural and other common caries risk factors, and caries development in permanent teeth is inconsistent. It is recommended that policy makers and healthcare professionals should dedicate oral health promotion and preventive strategies to children at a young age.

## Figures and Tables

**Figure 1 ijerph-19-13459-f001:**
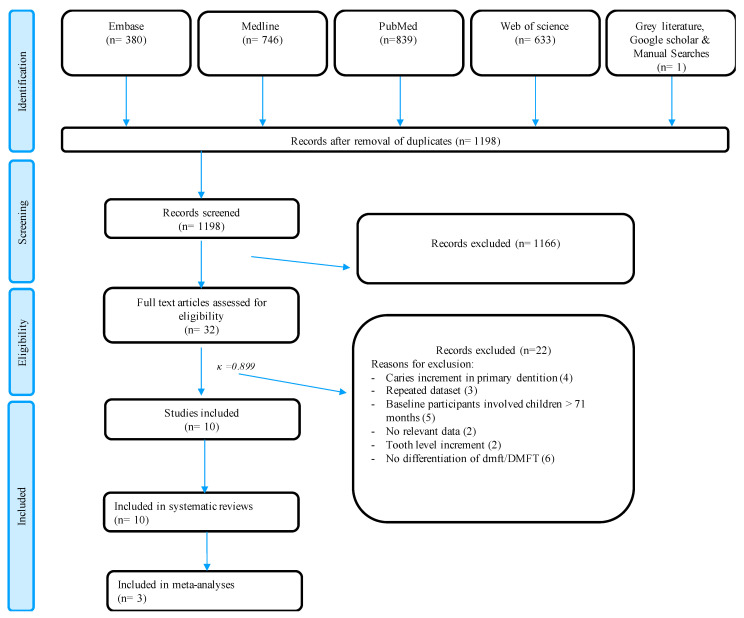
PRISMA flowchart of the current meta-evaluation.

**Figure 2 ijerph-19-13459-f002:**
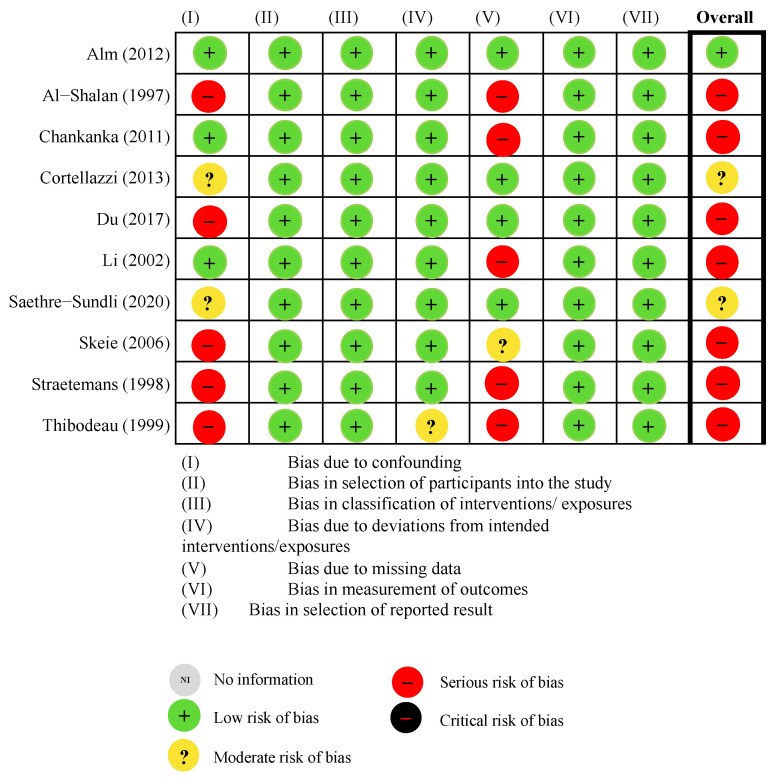
Risk of bias assessment.

**Figure 3 ijerph-19-13459-f003:**
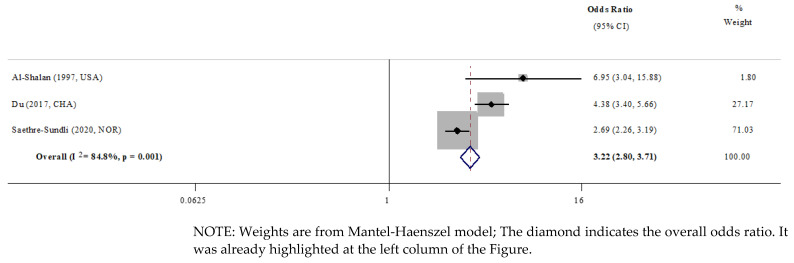
Meta-analysis. Presence of ECC [9,28,29].

**Figure 4 ijerph-19-13459-f004:**
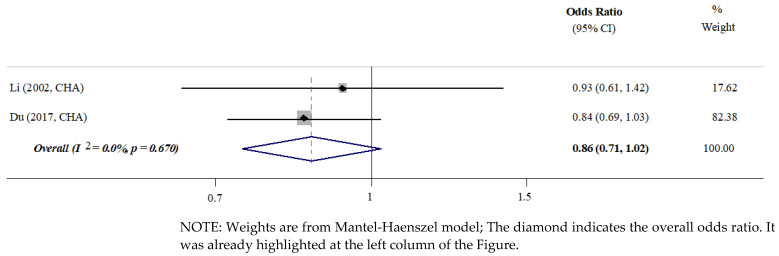
Meta-analysis. Gender [29,35].

**Table 1 ijerph-19-13459-t001:** Characteristics of included studies.

No.	Study (Year, Country ^a^)	Commencement Year, Duration	*N* Subject Baseline/ Follow-Up;	Commencement Age/Age at Final Follow-Up	Factors Evaluated	Measurement of Effect	Significant Factors
1	Alm (2012, SWE) [33]	1988, 14 years	671/568	12 months/ 15 years	- Gender - Oral hygiene status - Parental factors - Past caries experience - Snacking habit - Oral hygiene habit	- OR	- Caries experience at 6 years
2	Al-Shalan (1997, USA) [28]	1985, 3 years	115/115	18–48 months/ 6–14 years	- Gender - Age - cert/CERT - Caries location	- OR	- Caries in primary incisors - Sealants
3	Chankanka (2011, THA) [34]	1992, 5 years	198/150	5 years/9 years	- Gender - Dietary habit - Oral hygiene habit - Water fluoridation - Past caries experience	- OR	- Cavitated caries experience at 5 years - Less frequent toothburshing
4	Cortellazzi (2013, BRA) [8]	2005, 5 years	427/246	3–5 years/ 10 years	- Gender - Past DMFT - Gingivitis - Family income - No. household - Paternal & maternal education - Home and car ownership - Fluorosis	- DMFT prevalence - OR	- Past caries experience (DMFT)
5	Du (2017, CHA) [29]	2006 7 years	1885/1683	5 years/ 12 years	- Gender - Past DMFT score	- DMFT prevalence - OR	- Past caries experience (DMFT)
6	Li (2002, CHA) [35]	1992, 8 years	504/362	3–4 years/ 11–13 years	- Gender - Age - Socioeconomic factors - Past DMFT score	- Mean DMFT score - RR	- Past caries experience (DMFT) - Caries in primary molars
7	Saethre-Sundli (2020, NOR) [9]	2007, 7 years	3282/3282	5 years/ 12 years	- Approximal caries - Enamel caries - Dentinal caries - Parental ethnics - Parental education - Family status - Gender	- Mean DMFT score - OR	- Approximal caries - Enamel caries - Dentinal caries - Parental ethnics - Parental education - Family status
8	Skeie (2006, NOR) [30]	1993, 5 years	217/186	5 years/ 10 years	- Caries location	- OR	- Caries in primary second molars - Multi-surface caries
9	Straetemans (1998, NLD) [31]	1985, 6 years	196/109	5 years/ 11 years	- Mean & median mfs score - Mutans Streptococcus before & after 5 years old	- Mean & median MFS - Caries increment	- Mutans Streptococcus before 5 years old
10	Thibodeau (1999, USA) [32]	NR, 6 years	85/85	3 years/ 9 years	- Salivary mutans streptococci level	- Mean DMFS	- Salivary mutans streptococci level

^a^ ISO alpha-3 codes of Countries; cert/CERT: total number of carious, extracted and restored teeth; DMFT/S: number of decayed, missing or filled teeth/surface in permanent dentition; DMFT/s: number of decay, missing or filled teeth/surface in primary dentition; primary/secondary; OR: odds ratio; RR: risk ratio.

**Table 2 ijerph-19-13459-t002:** GRADE summary of findings.

Comparison	Results	*N* Patient Age Range (Years)	*N* Studies	Risk of Bias ^b^	Inconsistency ^c^		Indirectness ^d^	Imprecision ^e^	Publication Bias ^f^	Quality of Evidence (GRADE)
					I^2^ (%)	HETEROGENICITY χ^2^ Test (*p* Value)				
Past caries experience	Children with ECC are 3.22 times as likely to develop caries in permanent teeth	5080 (6–12)	3	Serious	84.8%	0.001	Not serious	Not serious	N/A	⊕OOO very low
				↓	↓		--	--	--	
Gender	No significant difference	2045 (11–13)	2	Serious	0.0%	0.670	Not serious	Not serious	N/A	⊕⊕OO low
				↓	--		--	--	--	

^b^ Risk of bias: Considered serious if half of the studies included were of serious risk of overall bias. ^c^ Inconsistency: Considered serious when I^2^ statistics ≥ 70% and *p*-value of χ^2^ test < 0.05 ^d^ Indirectness: Considered serious when applicability of findings were restricted in terms of population, intervention, comparator and outcomes. ^e^ Imprecision: Considered as serious when total number of events was below 300 for dichotomous outcomes or 400 for continuous outcomes, or when the upper and lower limits of 95% CI include both meaningful benefits and harm. ^f^ Publications bias: Considered serious if *p*-value of Begg’s funnel plot < 0.05. Not applicable (N/A) if funnel plot could not be constricted given limited numbers of study. Publication bias was difficult to detect, and thus no downgrading was performed. ↓: Downgrade by one level in quality of evidence; --: No change in quality of evidence.

## Data Availability

The data presented in this study are available in the Appendix A.

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
