# Peer review of "Does Early Childhood Caries Increase Caries Development among School Children and Adolescents? A Systematic Review and Meta-Analysis"

_ijerph, 2022, doi:10.3390/ijerph192013459_

Round 1

Reviewer 1 Report

This manuscript entitled “Does early childhood caries increase caries development 2 among school children and adolescents?” Is well written. However there are few shortcomings that need to be addressed.

Abstract: “four electronic databases” while in methods they are five. Please, be consistent.

Key words: HIV and antiretroviral therapy are not related to the topic of the review.

The aim does not match the title. In aim “caries in primary dentition” does not mean ECC. Please, be specific.

Results:

Fig 1: it does not include all the five databases included in the literature search while it includes PubMed that is not written in the methods.

“Meta-analyses from three studies”: more explanation is needed why only these studies were included in the meta-analysis.

Reviewer 2 Report

The manuscript entitled " Does early childhood caries increase caries development among school children and adolescents?" is a systematic review and meta-analysis regarding early childhood caries development and whether this aspect influences the number of caries at a bigger age (scholars and adolescents). The study itself is well described. The research confirms what other studies have shown regarding the association between early childhood caries and  However, some issues should be addressed:

- the title should be rephrased (the term"caries" is repeated & it should emphasize that it is a systematic review&meta-analysis);

- the abstract should be rephrased - the first sentence is unclear;

- it is unclear why "HIV" and "antiretroviral therapy" are keywords for this manuscript

Reviewer 3 Report

Although you well described the caries risk in children and aldults, there was lot of a bias in the various studies. E.g., the studies included in the meta-analysis all suffered from bias due to confounding and severe overall bias. Therefore, the data show in the meta-analysis has to be shown with much caution or the best is to omit them. With regard to the descriptive studies, in the discussion is the bias briefly mentioned while this is the major drawback of your study, it should be emphasized and also included in the conclusion. In the end, not a systemathic approach might be suitable for this study, but a narrative approach.

Round 2

Reviewer 3 Report

The paper has significantly improved. I would mention already in the results (paragraph 3.3)  that the high risk of bias results in the fact that the results of this study has to be interpreted with much caution.
